# DIVE DEEPER INTO INTEGRAL POSE REGRESSION

**Kerui Gu**[1]**, Linlin Yang**[1,2] **& Angela Yao**[1]
[1]National University of Singapore, Singapore
[2]University of Bonn, Germany
`{keruigu, yangll, ayao}@comp.nus.edu.sg`

## ABSTRACT

Integral pose regression combines an implicit heatmap with end-to-end training for human body and hand pose estimation. Unlike detection-based heatmap methods, which decode final joint positions from the heatmap with a non-differentiable argmax operation, integral regression methods apply a differentiable expectation operation. This paper offers a deep dive into the inference and back-propagation of integral pose regression to better understand the differences in performance and training compared to detection-based methods. For inference, we give theoretical support as to why expectation should always be better than the argmax operation, *i.e.* integral regression should always outperform detection. Yet, in practice, this is observed only in hard cases because the heatmap activation for regression shrinks in easy cases. We then experimentally show that activation shrinkage is one of the leading causes for integral regression's inferior performance. For back-propagation, we theoretically and empirically analyze the gradients to explain the slow training speed of integral regression. Based on these findings, we incorporate the supervision of a spatial prior to speed up training and improve performance.

## 1 INTRODUCTION

2D human pose estimation aims to detect the image coordinates of the body and/or the hand. In recent years, detection-based methods (Newell et al., 2016; Xiao et al., 2018; Sun et al., 2019; Li et al., 2019) and integral pose regression methods (Sun et al., 2018; Iqbal et al., 2018) have emerged as two common paradigms for human pose estimation. Both methods learn a heatmap representing pixel-wise likelihoods of the joint positions. The heatmap is learned explicitly for detection-based methods but remains implicit or "latent" for integral regression methods. To decode the heatmaps to joint coordinates, detection methods use an argmax, while integral regression methods take the expected value. As the expectation operation is differentiable, integral regression has the benefit of being end-to-end learnable, even though detection methods seem more competitive accuracy-wise (COC, 2020).

Gu et al. (2021)'s recent work showed a curious performance difference between integral regression and detection methods. With a nuanced split over the evaluation set, integral regression outperforms detection when test samples are "harder", *i.e.* with fewer keypoints present in the scene, under higher occlusion and lower bounding box resolutions. Given that both detection and integral regression methods work with fully convolutional feed-forward architectures, the questions naturally arise: What are the reasons behind this performance difference? Why is integral regression able to excel in these hard cases, when its overall performance seems to lag behind detection methods (COC, 2020)? These questions serve as the motivation for our closer study and analysis of integral pose regression.

Detection and integral regression methods differ in two aspects in both the forward and backward pass (see Fig. 1 for an overview). In the forward pass, the heatmap is decoded with an argmax for detection whereas a softmax normalization and expectation are used for integral regression. In the backward pass, detection methods are supervised with an explicitly defined Gaussian heatmap centered on the ground truth joint, while integral regression is supervised by the joint coordinates directly. Through detailed theoretical analysis and experimentation on the decoding and back-propagation process, we make the following findings and contributions:

1. We propose a unified model of the heatmap to interpret and compare detection and integral regression methods. We verify the model experimentally and show that as samples shift from hard to easy, the activation region on the heatmap shrinks for both detection and integral regression methods.

2. We demonstrate experimentally that degenerately small regions of activation degrade the accuracy of both detection and integral regression methods.

3. Integral regression methods, which decode the heatmap with an expectation operation, should result in a lower expected end-point error than detection methods that decode with an argmax operation. In practice, this can be observed only for hard samples due to the shrinkage of the active region on the heatmap.

4. Direct supervision with the joint coordinates in integral regression, although end-to-end, suffers from gradient vanishing and provides less spatial cues for learning the heatmap than the explicit heatmap supervision of detection methods. As a result, the training of integral regression is more inefficient and slower to converge than detection methods.

Our findings provide insight into integral pose regression, which has better theoretical performance, and show that the density of heatmaps plays an important role in decoding heatmaps to coordinates.

## 2 RELATED WORK

Since the concept of heatmaps were proposed in (Tompson et al., 2014), detection-based methods have been top performers in human pose estimation. Existing detection-based works emphasize extracting high-quality multi-resolution features. Particularly, Xiao et al. (2018) proposed to adapt ResNet with deconvolution layers while Hourglass (Newell et al., 2016) and Cascaded Pyramid Network (CPN) (Li et al., 2019) introduced cascaded network architectures with a coarse-to-fine design paradigm. High-Resolution Network (HRNet) (Sun et al., 2019) follows the coarse-to-fine paradigm and further improves performance by adopting more dense connections across different resolution representations. The integral regression method of Sun et al. (2018) introduced a competitive regression-based framework but is surpassed by more advanced detection-based works (Xiao et al., 2018; Sun et al., 2019).

Numerical regression-based methods, which directly regress joint coordinates, are commonly used in facial landmark detection (Feng et al., 2018; Zhu et al., 2020). These methods, however, are not as accurate as detection-based methods on human pose estimation. In order to improve the accuracy, integral regression methods attempt to implicitly merge the spatial knowledge from the heatmap via a "latent" heatmap. Integral regression methods are especially preferred in hand pose estimation (Spurr et al., 2020; Yang et al., 2021) but are still less common for human pose estimation (Sun et al., 2018; Nibali et al., 2018).

Recently, two parallel lines of work have emerged to analyse detection and integral regression methods separately. For detection-based methods, Huang et al. (2020) identified a heatmap bias caused by the inconsistency of the coordinate system transformations and solved it by redesigning the transformations during data processing. Additionally, as the predicted heatmaps during inference may violate the Gaussian assumption and worsen the performance, Huang et al. (2020) proposed to find the optimal Gaussian location based on the prediction while Zhang et al. (2020) modulated the predicted Gaussian distribution for decoding.

For integral regression methods, Nibali et al. (2018) experimentally compared different heatmap regularizers, heatmap normalization schemes and loss functions. They found that using Jensen-Shannon regularization on the heatmap with a softmax normalization and L1 loss achieves the best performance for integral regression. More recently, Gu et al. (2021) performed a systematic comparison of detection and integral regression methods with a common backbone and discovered the performance advantage of integral regression on the "hard" samples. Furthermore, Gu et al. (2021) revealed a bias that arises from taking the expectation after the softmax normalization. As such, they proposed a compensation scheme to mitigate the bias, which in turn improves the overall performance of integral regression methods, making them competitive with detection methods.

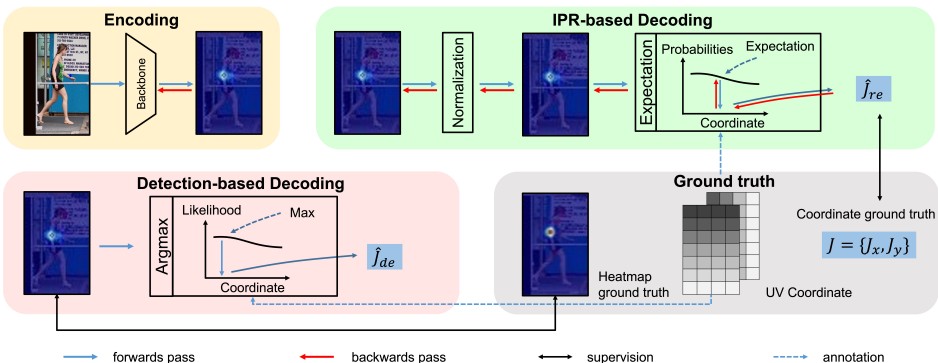

Figure 1: Comparison of the two decoding processes for detection-based methods (pink panel) and integral regression methods (green panel). They share the same encoding (yellow panel) but have different representations of the ground truth (gray panel).

## 3  PRELIMINARIES ON HUMAN POSE ESTIMATION

In this work, we target the more commonly used *'top-down'* form of pose estimation in which a person detector already provides a cropped image $I$ of the person. For simplicity, we focus our discussion on one given joint out of the $K$ total joints in the body. The pose estimation model outputs a heatmap $\hat{\mathbf{H}} \in \mathbb{R}^{M \times N}$, where $M$ and $N$ are the dimensions of the spatial heatmap. Typically, $M$ and $N$ are scaled down by a factor of $4$ from the original input dimensions of $I$ (Xiao et al., 2018; Sun et al., 2019). The heatmap $\hat{\mathbf{H}}$ represents a (discrete) spatial likelihood $P(\mathbf{J}|I)$, where $\mathbf{J} \in \mathbb{R}^{1 \times 2}$ is the 2D coordinates of the joint. In practice, all $K$ heatmaps are predicted simultaneously by the same network, where each joint is one channel. In both detection and integral regression methods, the coordinates $\hat{\mathbf{J}}$ are decoded from $\hat{\mathbf{H}}$, where the two methods differ in the manner of decoding (see Sec. 3.1) and the form of supervision applied (see Sec. 3.2).

### 3.1  HEATMAP DECODING: MAX VERSUS EXPECTED VALUE

**Detection methods** apply an argmax on $\hat{\mathbf{H}}$ indexed by $\mathbf{p}$ to estimate the joint coordinates $\hat{\mathbf{J}}_{\text{de}}$:

$$\hat{\mathbf{J}}_{\text{de}} = \arg \max_{\mathbf{p}} \hat{\mathbf{H}}(\mathbf{p}). \tag{1}$$

Taking an argmax can be interpreted as taking a maximum likelihood on the heatmap $\hat{\mathbf{H}}$, assuming that $\hat{\mathbf{H}}$ is proportional to the likelihood. In practice, the final $\hat{\mathbf{J}}_{\text{de}}$ value is determined as a linear combination of the highest and second-highest response on $\hat{\mathbf{H}}$ as a way to account for quantization effects in the discrete heatmap (Newell et al., 2016). A more recent work, DARK (Zhang et al., 2020), approximates the true prediction by a Taylor series evaluated at the maximum activation of the heatmap and shows this to be more accurate.

**Integral pose regression** applies an expectation operation on $\hat{\mathbf{H}}$ to give a mean estimate of the joint coordinates. To do so, the heatmap must first be normalized to sum up to 1; the most common and effective approach (Nibali et al., 2018) is to apply a softmax normalization. Afterwards, the predicted joint $\hat{\mathbf{J}}_{\text{re}}$ with $x$ and $y$ components $\hat{J}_x$ and $\hat{J}_y$[1] is determined by taking the expectation on the normalized heatmap $\tilde{\mathbf{H}}$ with elements $\tilde{h}_{\mathbf{p}}$ at location $\mathbf{p}$:

$$\hat{\mathbf{J}}_{\text{re}} = \begin{bmatrix} \hat{J}_x \\ \hat{J}_y \end{bmatrix} = \sum_{\mathbf{p} \in \Omega} \mathbf{p} \cdot \tilde{h}_{\mathbf{p}} \qquad \text{where} \qquad \tilde{h}_{\mathbf{p}} = \frac{e^{\beta \hat{h}_{\mathbf{p}}}}{\sum\limits_{(\mathbf{p}') \in \Omega} e^{\beta \hat{h}_{\mathbf{p}'}}}. \tag{2}$$

Here, $\Omega$ is the domain of the heatmap and $\beta$ is a scaling parameter used in the softmax normalization. Note that softmax normalization assigns a non-zero value to *all* pixels in $\tilde{\mathbf{H}}$, even if it was originally zero in $\hat{\mathbf{H}}$. These values also contribute to the expected value, resulting in a center-biased estimated

---

[1]For clarity, we drop the subscript 're', as we refer to the individual components for integral regression only.

joint coordinate $\hat{J}_{re}$ (Gu et al., 2021). The smaller the $\beta$, the stronger the bias. Although Gu et al. (2021) proposed a compensation scheme, for the purpose of our analysis, we will assume that $\beta$ is sufficiently large such that the impact of the bias is negligible.

## 3.2 SUPERVISION: EXPLICIT HEATMAP VERSUS GROUND TRUTH COORDINATES

**Detection methods** are learned by providing supervision on the heatmap. The ground truth $\mathbf{H}$ is given as a circular Gaussian, with a mean centered at the ground truth joint coordinate (see Fig. 1). The loss applied is a pixel-wise MSE between the ground truth $h_{\mathbf{p}}$ and the predicted $\hat{h}_{\mathbf{p}}$:

$$L_{de} = ||\mathbf{H} - \hat{\mathbf{H}}||_2^2 = \sum_{\mathbf{p} \in \Omega} (h_{\mathbf{p}} - \hat{h}_{\mathbf{p}})^2, \tag{3}$$

where $\Omega$ is the domain of the heatmap. As the loss is defined in terms of the heatmap and not in terms of the predicted joint coordinates, detection-based methods are not end-to-end in their learning, and this is often cited as a drawback (Sun et al., 2018; Zhang et al., 2020).

**Integral pose regression** defines a loss based on the difference between the prediction $\hat{\mathbf{J}}_{re}$ and the ground truth joint location $\mathbf{J}$. The L1 loss empirically performs better than L2 (Sun et al., 2018):

$$L_{re} = ||\mathbf{J}_{gt} - \hat{\mathbf{J}}_{re}||_1 = (|\hat{J}_x - J_x| + |\hat{J}_y - J_y|). \tag{4}$$

Integral regression methods are end-to-end because they provide supervision at the joint level. As $\hat{\mathbf{H}}$ is learned only implicitly, some works refer to the heatmap as "latent" (Iqbal et al., 2018).

## 3.3 PERFORMANCE DIFFERENCES

Sun et al. (2018) and Gu et al. (2021) compared the performance of detection and integral regression methods on human pose estimation. Sun et al. (2018) found that integral regression is either as competitive or better for 2D pose estimation, though results are not fully conclusive, as the backbone they used for integral regression is different from detection.

The more systematic and detailed comparison from Gu et al. (2021) used a common backbone. It also partitioned the data based on three factors of variation dominating current human pose estimation benchmarks (Ruggero Ronchi & Perona, 2017): the number of joints present in the scene, the percentage of occlusion and the largest dimension of the bounding box input. The *"hard"* samples are those with either 1-5 joints present, $> 50\%$ occlusion, or 32-64px input. For these samples, integral regression had, on average, 15% lower end-point error (EPE) than detection. The *"easy"* samples are defined by 11-17 joints, $<10\%$ occlusion, or $>128$px input. In these cases, detection methods were marginally better. The remaining cases with factors in between were considered to be *"medium"*; performance of both methods was roughly equivalent on these samples.

## 4 ANALYSIS ON HEATMAP DECODING

### 4.1 LOCALIZED HEATMAP MODEL

To analyze the performance differences between detection and integral regression methods, we make the following assumptions on the heatmap itself. First and foremost, we assume that a well-trained network will produce a heatmap $\hat{h}_{\mathbf{p}}$ with significant or large values in a *localized* support region around $\hat{\mathbf{J}}$. We denote the region of support as $\Phi$ and assume that outside of $\Phi$, the heatmap activation is approximately zero (see Fig. 2 (a)).

Specifically, the normalized heatmap[2] $\tilde{\mathbf{H}}$ is modelled by some density distribution $\mathcal{P}$. We make no assumption on the form of $\mathcal{P}$, but do assume that it fully captures the support for the joint location within $\Phi$ and that $\Phi$ is centered on the expected value of $\mathcal{P}$ on location $\boldsymbol{\mu} = (\mu_x, \mu_y)$ with an area of $(2s + 1, 2s + 1)$, *i.e.*

$$\boldsymbol{\mu} = \sum_{\mathbf{p} \in \Omega} \mathbf{p} \cdot \tilde{h}_{\mathbf{p}} = \sum_{\mathbf{p} \in \Phi(\boldsymbol{\mu}, s)} \mathbf{p} \cdot \tilde{h}_{\mathbf{p}} = \left[ \sum_{i=\mu_x-s}^{\mu_x+s} \sum_{j=\mu_y-s}^{\mu_y+s} i \cdot \tilde{h}_{ij}, \sum_{i=\mu_x-s}^{\mu_x+s} \sum_{j=\mu_y-s}^{\mu_y+s} j \cdot \tilde{h}_{ij} \right]^{\mathsf{T}}. \tag{5}$$

---

[2]For the purpose of discussion, we may consider normalizing the heatmap for detection methods as per Eq. 2, which will not affect the outcome since $\arg\max(\tilde{\mathbf{H}}) = \arg\max(\hat{\mathbf{H}})$.

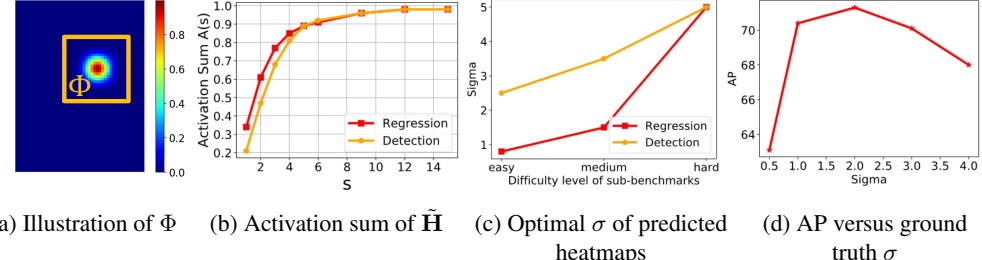

(a) Illustration of $\Phi$    (b) Activation sum of $\tilde{\mathbf{H}}$    (c) Optimal $\sigma$ of predicted heatmaps    (d) AP versus ground truth $\sigma$

Figure 2: (a) The heatmap is only activated within $\Phi$; (b) more heatmap activations are bounded by a $(2s+1) \times (2s+1)$ region for integral regression than detection; (c) if the heatmap is approximated as a Gaussian, the resulting $\sigma$ shrinks as samples shift from hard to easy; this shrinkage is faster and more extreme for integral regression compared to detection; (d) in detect methods, the optimal $\sigma$ is around 2; smaller and larger $\sigma$ hurt performance.

**Experimental Verification of Localized Heatmap Model**   The assumption of localized activations is reasonable for detection methods as their heatmaps are learned explicitly to match a (localized) ground truth Gaussian. We empirically verify this model by tallying the activations in the normalized heatmap for both detection and integral regression methods based on the easy/medium/hard splits as specified in Sec. 3.3.

Specifically, we train both a detection and an integral regression network for human pose estimation. For both networks, we use a Simple Baseline (SBL) (Xiao et al., 2018) architecture with a ResNet50 (He et al., 2016) backbone. The two networks differ only in their manner of decoding and form of supervision as outlined in Section 3. We then apply the networks on the standard human pose estimation benchmark MSCOCO (Lin et al., 2014) using the same easy, medium and hard splits as Gu et al. (2021). For each produced heatmap of $64 \times 48$, we estimate an activation sum $A$ by summing the normalized heatmap activations within a $(2s+1) \times (2s+1)$ square around the ground truth coordinate, $(J_x, J_y)$:

$$A(s) = \sum_{i=J_x-s}^{J_x+s} \sum_{j=J_y-s}^{J_y+s} \tilde{h}_{ij}. \tag{6}$$

From Fig. 2 (b), we show that with a sufficiently large s, *e.g.* 9, the activation sum exceeds 95%.

## 4.2 Extreme Localization as a Cause for Performance Differences

**Expected End-Point Error (EPE)**   For detection methods, which decode the heatmap with an argmax, we assume that each position $\mathbf{p}$ has some probability $w(\boldsymbol{\mu}, s)$ of being the maximum activation, *i.e.* $\arg\max(\mathcal{P})_{\Phi(\mu,s)} \sim w(\mu, s)$. Here, $w(\mu, s)$ represents a radially symmetric distribution centered on $\boldsymbol{\mu}$ with all non-zero support contained within $\Phi$. It follows that the expected EPE for one sample of a detection method can be defined as:

$$\mathbb{E}_{\mathrm{de}}(\boldsymbol{\mu}, s) = \sum_{\mathbf{p} \in \Phi(\boldsymbol{\mu}, s)} w(\mathbf{p}) \|\mathbf{J}_{\mathrm{gt}} - \mathbf{p}\|_2, \tag{7}$$

where $\mathbf{J}_{\mathrm{gt}}$ denotes the ground truth coordinates. For integral regression methods, which decode the heatmap with an expectation, the estimated joint coordinate aligns with the center of $\Phi$ by definition, *i.e.* $\hat{\mathbf{J}} = \boldsymbol{\mu}$, leading to the following expected EPE for a single sample:

$$\mathbb{E}_{\mathrm{re}}(\boldsymbol{\mu}) = \|\mathbf{J}_{\mathrm{gt}} - \boldsymbol{\mu}\|_2. \tag{8}$$

It can be verified that the $\mathbb{E}_{\mathrm{de}} \geq \mathbb{E}_{\mathrm{re}}$ (see proof in Appendix A.2). This result suggests that for some fixed $(\boldsymbol{\mu}, s)$, the expected EPE of regression should always be better than detection. However, the results of Gu et al. (2021) as summarized in Sec. 3.3 have clearly shown otherwise. This inconsistency can be better understood with the following two experiments.

**Experimental Verification: Integral Regression Methods Are "More" Localized than Detection**   The plot in Fig. 2 (b) indicates that regression methods seem to be *more* localized than detection methods. In particular, for smaller $s$, $A$ is smaller for detection methods than integral regression methods.

| Model | Detection | Regression | $+\sigma=0.5$ | $+\sigma=1$ | $+\sigma=2$ | $+\sigma=3$ | $+\sigma=4$ |
|---|---|---|---|---|---|---|---|
| AP | 71.3 | 67.1 | 65.1 | 68.5 | 71.4 | 69.8 | 66.9 |
| $AP_{e/m}$ | 74.3 | 70.4 | 68.2 | 71.9 | 74.1 | 73.0 | 70.2 |
| $AP_h$ | 43.2 | 44.7 | 43.9 | 45.2 | 47.4 | 47.6 | 46.9 |
| $A(s=2)_{e/m}$ | 0.49 | 0.65 | 0.75 | 0.63 | 0.54 | 0.49 | 0.45 |
| $A(s=2)_h$ | 0.30 | 0.35 | 0.37 | 0.36 | 0.41 | 0.39 | 0.39 |

Table 1: Performance and activation sum of detection baseline, regression baseline, and regression baseline added KLDiv loss with different $\sigma$s. When $\sigma = 0.5, 1$, the results are not improved but even worsened. When $\sigma = 2$, the spread of heatmaps is smoothed and the results even exceed detection baselines.

To further verify, we compare the heatmaps with an idealized Gaussian heatmap of varying standard deviations or $\sigma$. We plot the $\sigma$ that ' gives the lowest Pearson Chi-square statistic, *i.e.* the highest similarity for the different levels of difficulty in Fig. 2 (c) (detailed values are provided in Appendix A.3). As expected, the optimal $\sigma$ decreases as the samples progress from hard to easy; this result is in line with Fig. 2 (b) and shows that heatmaps are more localized for easy samples. However, we also observe that the optimal $\sigma$ for intgral regression is much smaller for the medium and easy cases. We speculate that this difference arises because detection methods are trained with a Gaussian heatmap of $\sigma = 2$ (Xiao et al., 2018; Sun et al., 2019). Integral regression methods, on the other hand, have no such explicit supervision, and therefore no restriction on the extent of localization.

**Experimental Verification: Extremely Localized Heatmaps Degrade Performance**   Is it possible that smaller hypothetical $\sigma$, *i.e.* more localized heatmaps, causes poor performance? We verify this for detection methods by applying ground truth Gaussian heatmaps of varying $\sigma$ for training the networks as per Sec. 4.1 and evaluate network performance using Average Precision (AP). A higher AP corresponds to more accurately located joints. Fig. 2 (d) confirms that $\sigma = 2$ is optimal. A smaller $\sigma = 1$ degrades the performance slightly, and there is a significant drop in the extreme case when $\sigma = 0.5$.

To verify the impact of $\sigma$ on regression methods, we add a distribution prior to the heatmap to prevent the shrink of the support region $\Phi$. Specifically, we apply a Kullback-Leibler Divergence (KLDiv) loss between the non-normalized heatmap $\hat{\mathbf{H}}$ and a Gaussian heatmap with varying $\sigma$. From Table 1, we see that the regression baseline is worse than detection baseline. However, when we add a prior on the heatmap distribution to encourage a sufficiently large $\Phi$, *i.e.* ($+\sigma = 1, 2, 3$), performance improves. At the optimal $+\sigma = 2$, integral regression becomes competitive with and even exceeds the detection baseline with the optimal $\sigma$. Most importantly, extreme $\sigma$ values cause performance to degenerate, especially in the case of the extremely small $+\sigma = 0.5$.

We also demonstrate the performance and localization extent of easy/medium and hard cases by AP and activation sum $A(s = 2)$. After adding priors (*e.g.* , $+\sigma = 2$), $A(s = 2)_{e/m}$ decreases from 0.65 to 0.54, indicating the heatmaps are less localized, and the performance $AP_{e/m}$ is significantly improved. We provide the detailed training information regarding different loss weights for better understanding in Appendix A.4 and detailed changes in heatmap shrinkage in Appendix A.5.

## 5   ANALYSIS OF SUPERVISION AND LEARNING

Given the same heatmap, different methods of heatmap decoding not only affect the final coordinates but also determine the gradients that arise from the loss. The gradients in turn influence the learning and thereby change the generation of the heatmaps. In this section, we explore the gradients of heatmaps with respect to the loss functions for detection and integral regression methods and pinpoint the specific gradient components that slow down the learning of integral regression.

## 5.1 HEATMAP GRADIENTS

**Detection Methods**  The gradient of $L_{\text{de}}$ (see Eq. 3) with respect to each pixel in the estimated heatmap $\hat{h}_{\mathbf{p}}$ is straightforward:

$$\frac{\partial L_{\text{de}}}{\partial \hat{h}_{\mathbf{p}}} = 2(\hat{h}_{\mathbf{p}} - h_{\mathbf{p}}). \tag{9}$$

The gradient in Eq. 9 features the predicted heatmap value $\hat{h}_{\mathbf{p}}$ and the ground truth heatmap value $h_{\mathbf{p}}$. It provides explicit supervision at every pixel. False positives *and* false negatives are penalized by reducing wrong high likelihoods and raising incorrect low likelihoods, respectively. For each position $\mathbf{p}$ in the heatmap, if the predicted $\hat{h}_{\mathbf{p}}$ is smaller than the ground truth $h_{\mathbf{p}}$, the gradient is negative and proportionally increases the value in the next iteration. The vice versa is true for the predicted $\hat{h}_{\mathbf{p}}$ larger than the ground truth.

**Integral Regression**  The gradient of the loss $L_{\text{re}}$ (see Eq. 4) with respect to $\hat{h}_{\mathbf{p}}$ can be estimated based on the chain rule as (see detailed derivation in Appendix A.6)

$$\nabla_{\mathbf{p}} := \frac{\partial L_{\text{re}}}{\partial \hat{h}_{\mathbf{p}}} = \frac{\partial L_{\text{re}}}{\partial \hat{\mathbf{J}}} \frac{\partial \hat{\mathbf{J}}}{\partial \tilde{h}_{\mathbf{p}}} \frac{\partial \tilde{h}_{\mathbf{p}}}{\partial \hat{h}_{\mathbf{p}}} = \beta \underbrace{\tilde{h}_{\mathbf{p}}}_{\nabla_1 \text{ (value factor)}} \underbrace{(\text{s}(\hat{J}_x - J_x)(i - \hat{J}_x) + \text{s}(\hat{J}_y - J_y)(j - \hat{J}_y))}_{\nabla_2 \text{ (location factor)}}. \tag{10}$$

In Eq. 10, the gradient is split into two terms of interest: $\nabla_1$ and $\nabla_2$, which we name as the value factor and the location factor, respectively. The nature of the value and location factors gives strong indication to the learning process for integral regression methods. Firstly, the gradient $\nabla_{\mathbf{p}}$ is proportional to the normalized predicted heatmap value $\tilde{h}_{\mathbf{p}}$ as given in the value factor. This factor makes it prone to gradient vanishing wherever locations $\mathbf{p}$ have small heatmap predictions. Secondly, the location factor is a linear combination of $i$ and $j$, and it manifests itself as a linear plane; far away points like corners are more likely to have large magnitudes. Though these two factors do not prevent the loss from decreasing during learning, they slow down training. We further elaborate this in Sec. 5.2.

## 5.2 A DETAILED ANALYSIS OF THE INTEGRAL REGRESSION HEATMAP

We now conduct a detailed analysis of the gradients presented in Eq. 10 and show the typical cases that arise during learning. When referring to $\Omega$, the domain of the heatmap, we use a coordinate system with the origin at the upper-left corner and increasing coordinates moving to the lower right corner (see Fig. 2 (a)). For simplicity of discussion, we assume that the ground truth joint $\mathbf{J}_{\text{gt}}$ is located somewhere in the lower-right quadrant of $\Omega$, though the analysis holds for $\mathbf{J}_{\text{gt}}$ in all other quadrants of $\Omega$ analogously. The four scenarios are visualized in Fig. 3.

We base our analysis on the assumption that a larger magnitude of $\nabla_{\mathbf{p}}$ has a larger influence on the weights of the network and causes a greater change at the location $\mathbf{p}$ in the next iteration, *i.e.*

$$h_{\mathbf{p}}^{n+1} \approx h_{\mathbf{p}}^n - \gamma \nabla_{\mathbf{p}}^n, \tag{11}$$

where $\gamma$ represents the step size or learning rate (see sketch of proof in Appendix A.7) and $n$ denotes the update iteration. As the value and location factors have different contributions on the gradient from Eq. 10, and they are in turn affected by the size $s$ and location $\mu$ of $\Phi$, we can define four characteristic cases of $\Phi$ and outline how learning is affected in each case.

**(1) $s$ is large, $\Phi$ has uniformly random values**, *e.g.* a randomly initialized heatmap at the start of the training. As $\tilde{h}_{\mathbf{p}}$ is in a similar range for all $\mathbf{p}$, the value factor $\nabla_1$ (see Eq. 10) of all pixels are approximately similar. As such, the distinction between one pixel's gradient and that of another is determined by the location factor $\nabla_2$. Given that the ground truth coordinates $(J_x, J_y)$ are in the lower-right quadrant, as long as $(J_x, J_y)$ are both greater than the predicted coordinate $(\hat{J}_x, \hat{J}_y)$, $\nabla_{\mathbf{p}}$ has a gradation that increases progressively towards the bottom right corner. Generally speaking, the location factor pushes the activations of the heatmap towards the corner of the correct quadrant as corners own the largest gradients in the resultant linear plane of the location factor $\nabla_2$.

**(2) $s$ is small, $\mu$ is far from the ground truth J**, *e.g.* when $\mu$ is in the upper left quadrant of $\Omega$ while the ground truth $\mathbf{J}_{\text{gt}}$ is at the bottom right. A typical scenario is if the heatmap activates around the

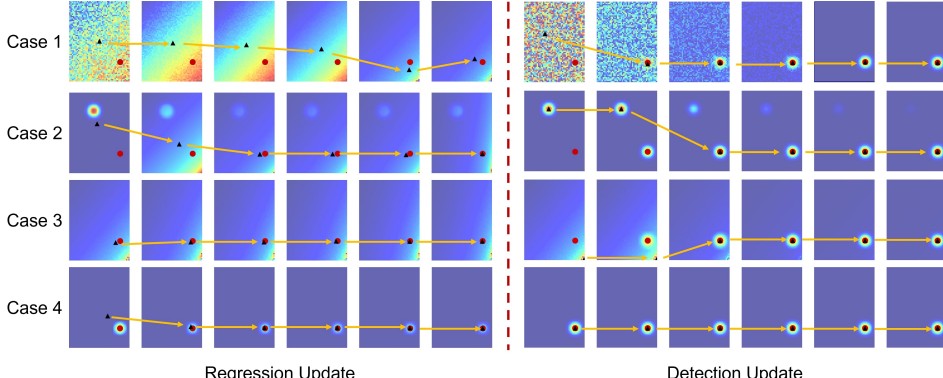

Figure 3: Toy example of regression update (left) and detection update (right). The black triangle denotes the prediction (expectation value for regression and max value for detection) while the red dot denotes the ground truth position. The arrow shows the movement of the prediction.

left ankle for the ground truth right ankle. In this case, for the pixels outside of $\Phi$, the value factor approaches zero, *i.e.* $\tilde{h}_{\mathbf{P}} \to 0$, and pushes $\nabla_{\mathbf{P}}$ towards zero, hence the pixels receive very limited gradient updates. For pixels in $\Phi$, the values decrease gradually until the value factor outside of $\Phi$ no longer dominates, *i.e.* they reach the same scale, at which point the heatmap returns to case (1).

**(3) $s$ is small, $\mu$ is in the corner of the same quadrant as the ground truth $\mathbf{J}_{\mathbf{gt}}$.** Initially, this case is similar to case (2). In the process of all the elements becoming the same scale, all the activations move towards the diagonal side and the prediction approaches the ground truth coordinate.

**(4) $s$ is small, $\mu$ close to ground truth $\mathbf{J}_{\mathbf{gt}}$.** This scenario occurs when the model is reasonably trained and can roughly localize the joint. The gradient at the ground truth pixel $\mathbf{J}_{\mathrm{gt}}$ is always negative or zero, *i.e.*

$$\nabla_{(J_x, J_y)} = -\beta \tilde{h}_{(J_x, J_y)} (\|\hat{J}_x - J_x\| + \|\hat{J}_y - J_y\|). \tag{12}$$

This non-positive value guides the network to predict a large heatmap response at the ground truth location, *i.e.* a large $\tilde{h}_{(J_x, J_y)}$; this in turn increases the gradient $\nabla_{(J_x, J_y)}$. This property makes the network more likely to predict a few exceptionally large pixels, thus shrinking the support region $\Phi$ and leading to a very small $\sigma$ in Sec. 4.1.

For detection-based methods, the ground truth heatmaps have already explicitly pointed out whether or not a pixel should be activated in the heatmap. Combined with its effective pixel-wise loss, the network learns faster than regression-based methods; this is validated in the next section.

## 5.3 EXPERIMENTAL VERIFICATION

**Idealized Sample** We start by considering the case of one sample for a single joint and visualize the progression of a $(64, 48)$ heatmap as it gets updated by Eq. 11. While $\nabla_{\mathbf{P}}^n$ can be calculated mathematically by Eq. 10, we use autograd in Pytorch to obtain the gradients, which we verify to be equivalent. The four cases from Sec. 5.2 are initialized randomly, with a symmetric Gaussian ($\sigma = 2$) centered at some upper left point, with a linear plane at the lower right quadrant, and a symmetric Gaussian ($\sigma = 2$) centered at ground truth, respectively. The progression of the differently initialized heatmaps can be observed in Fig. 3.

Mathematically, it is clear from Eq. 2 that the resulting joint coordinate location from integral regression can align with the ground truth joint even if the underlying heatmap does not follow a localized heatmap model. This is also illustrated in the first three cases discussed in Sec. 5.2. However, from the thousands of training samples observed over many epochs, we posit that during real-world training, the network can only consistently lower the loss if the heatmap $\tilde{H}$ is learned to represent $P(\mathbf{J}|I)$. As such, the activation region $\Phi$ will be correctly localized over the corresponding semantic region in the image. Instead, what slows down convergence is the learning of $\Phi$, or rather, the lack of direct guidance to yield a correct $\Phi$ as per detection methods. For the same four cases, we visualize the updated heatmap of Eq. 11 from the gradients defined by the detection loss in Eq. 9.

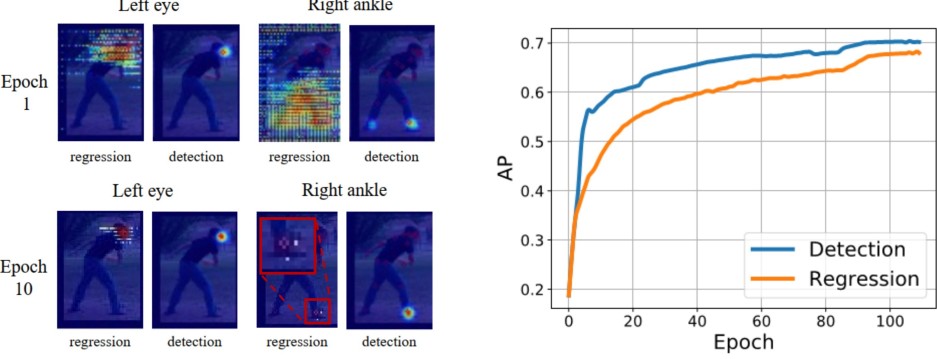

(a) Comparisons of detection and regression in real training

(b) Training Efficiency of Detection versus Regression on COCO validation set

Figure 4: (a) Comparisons of detection and regression in real training on an image selected from the COCO validation set at epoch 1 and 10. Results of detection have already roughly stabilized while regression can only localize to a small extent in epoch 10 with some background pixels still activated. We enlarge the right ankle part of regression in the red box for better visualization. More examples are shown in Appendix A.8. (b) The regression method converges slower than the detection method.

Unlike regression, each case leads consistently to the same localized $\Phi$ centered on the ground truth joint coordinate, regardless of the initialization.

**Real World Samples**   The four cases listed in Sec. 5.2 are not so clearly observable in real-world training as the output is based on the optimization results of batch-wise training in a real network. To verify the results, we compare the training progression for a detection and an integral regression network under the experimental settings described in Sec. 4.1. Fig. 4 (b) compares the training speed of both detection and integral regression and shows the slower convergence of integral regression.

We visualize for one sample from the MSCOCO validation set the heatmap at epochs 1 and 10 to compare the training progression for detection and integral regression. Fig. 4 (a) shows the heatmaps of the "left eye" and "right ankle" joints. For regression, after one epoch, the activations are spread over a quarter to half of the heatmap. This aligns roughly to a mix of Case 1 and 3 where activations are in the correct quadrant but not yet localized to a small region of support. After 10 epochs of training, predictions for the left eye (which is actually occluded, making it a "hard" sample) are localized with several activations of approximately the same value (all shown as red). For the right ankle (which is clearly present and is an "easy" sample), the network predicts heatmap values that have collapsed spatially and dominate at one or two pixel locations.

In contrast, the detection method presents a well-localized heatmap after only one epoch of training. While there is some activation on the left ankle in epoch 1, this is unlikely to cause errors for the argmax decoding. After 10 epochs, all activations are centered around the correct right ankle.

## 6    CONCLUSION

In this paper, we dive deeper into integral pose regression by comparing the differences in the heatmap activations and the gradients between detection and integral regression methods. Our theoretical analysis paired with empirical verification shows that the shrinkage of heatmaps of the integral regression method is the main cause for its lower performance compared to detection. Furthermore, we investigate the gradients of integral regression by giving theoretical evidence, toy problem and real world illustrations to show that the learning of integral regression is slower to converge as a result of only implicit supervisory signals. To mitigate the slow convergence and poor performance, we propose a simple spatial supervision on top of the existing framework.

**Acknowledgements**   This research / project is supported by the Ministry of Education, Singapore, under its MOE Academic Research Fund Tier 2 (STEM RIE2025 MOE-T2EP20220-0015).

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

# A Appendix

## A.1 Extension of Fig. 2 (b)

We provide an extended plot of Fig. 2 (b), which includes both detection and regression on easy, medium, and hard cases in Fig. I.

## A.2 Proof for Eq. 7 ⩾ Eq. 8 in Sec. 4.2

This subsection gives the detailed proof for Eq. 7>Eq. 8, which means

$$\sum_{\mathbf{p}\in\Phi(\boldsymbol{\mu},s)} w(\mathbf{p})||\mathbf{J}_{\text{gt}} - \mathbf{p}||_2 \geqslant ||\mathbf{J}_{\text{gt}} - \boldsymbol{\mu}||_2, \text{where} \sum_{\mathbf{p}\in\Phi(\boldsymbol{\mu},s)} w(\mathbf{p}) = 1. \tag{I}$$

In this way, this equation can be rewritten as

$$\sum_{\mathbf{p}\in\Phi(\boldsymbol{\mu},s)} w(\mathbf{p})||\mathbf{J}_{\text{gt}} - \mathbf{p}||_2 \geqslant \sum_{\mathbf{p}\in\Phi(\boldsymbol{\mu},s)} w(\mathbf{p})||\mathbf{J}_{\text{gt}} - \boldsymbol{\mu}||_2. \tag{II}$$

As the distribution of $\Phi(\boldsymbol{\mu}, s)$ is radially symmetric, a sufficient but unnecessary proof is that for any symmetric pair $\{\mathbf{p}_1(\mu_x - a, \mu_y - b), \mathbf{p}_2(\mu_x + a, \mu_y + b)\}$, it satisfies

$$\sqrt{(x - \mu_x - a)^2 + (y - \mu_y - b)^2} + \sqrt{(x - \mu_x + a)^2 + (y - \mu_y + b)^2} \geqslant 2\sqrt{(x - \mu_x)^2 + (y - \mu_y)^2}. \tag{III}$$

According to Triangle Inequality in normed vector space, we arrive at

$$||\mathbf{u}||_2 + ||\mathbf{v}||_2 \geqslant ||\mathbf{u} + \mathbf{v}||_2. \tag{IV}$$

When we set $\mathbf{u} = (x - \mu_x - a, y - \mu_y - b)$ and $\mathbf{v} = (x - \mu_x + a, y - \mu_y + b)$, Eq. II is established.

## A.3 Pearson Chi-square values on different cases

We report the detailed Pearson Chi-square values between heatmaps of both detection and regression in different scenarios and Gaussian templates with different values of $\sigma$ in Table. I, II, III. We set the threshold of $A(s) > 0.8$ to show that the support region can represent the whole heatmaps. Therefore, for easy and medium cases, $s = 4$; while for hard cases, $s = 8$.

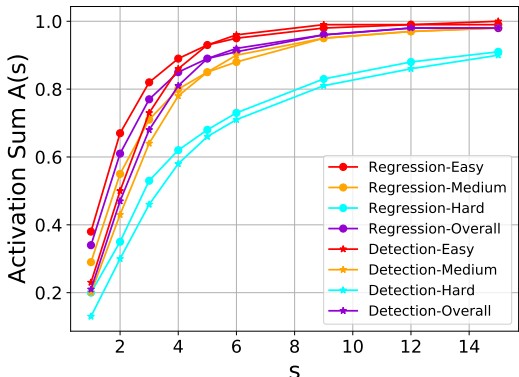

Figure I: Activation Sum $A(s)$ of both detection and regression method on easy, medium, hard, and overall cases.

| $\sigma$ | regression | | | | detection | | | |
|---|---|---|---|---|---|---|---|---|
| $s$ | 2 | 3 | 4 | 5 | 2 | 3 | 4 | 5 |
| 8 | >1000 | 4.76 | 4.12 | 3.95 | >1000 | 3.51 | 3.31 | 3.25 |

Table I: Results of the Pearson Chi-square test between distribution of heatmap $\mathbf{H}$ and Gaussian templates $\mathbf{T}$ with different window size $s$ and standard deviation $\sigma$ in hard cases. $\sigma = 5$ is the best-matching template for both detection and regression. $\sigma$ stops at 5 as the Gaussian templates with size 8 become similar after.

| $\sigma$ | regression | | | | detection | | | |
|---|---|---|---|---|---|---|---|---|
| $s$ | 1 | 2 | 3 | 4 | 1 | 2 | 3 | 4 |
| 4 | >1000 | 3.25 | 3.65 | 3.73 | >1000 | 1.76 | 0.76 | 0.77 |

Table II: Results of the Pearson Chi-square test between distribution of heatmap $\mathbf{H}$ and Gaussian templates $\mathbf{T}$ with different window size $s$ and standard deviation $\sigma$ in medium cases. For detection, best matched $\sigma$ should be between 3 and 4; while for regression, optimal $\sigma$ in medium case should be between 1 and 2.

## A.4 DETAILS OF COMBINING PRIOR LOSS AND INTEGRAL LOSS

This subsection demonstrates the influence of weights of prior loss and integral loss, and the magnitudes of two gradients regarding the shared heatmaps during the training.

Let us assume $w_p$ and $w_i$ are weights of the prior and integral losses, respectively. We consider the ratio $w_p L_p : w_i \cdot \lambda L_i$, where $\lambda = 10^{-2}$, to set the two to approximately equal magnitudes at the initial epoch. Table V shows that the AP heavily depends on the value of $\sigma$ but for some fixed $\sigma$ (within each row), the ratio of the loss weightings has less impact and the results are stable within $1 - 2\%$, with a larger influence only at the extreme settings of $100 : 1$ or $1 : 100$.

Table IV (b) provides the ratios of the gradient magnitudes to help explain why the weightings have little impact. Table IV (b) gives the mean ratio $\frac{g_p}{g_i}$ where $g_p$ and $g_i$ are the gradient magnitudes of the prior loss and integral loss, respectively, over the course of training. A column of 2/1/0/-1/-2 means $10^2/10^1/10^0/10^{-1}/10^{-2}$ ratio in gradients over the epochs. The table shows that regardless of the weighting, the gradient ratio decreases over the course of training (*i.e.* dropoff with each column). Apart from an extremely high weighting of the prior, *i.e.* 100:1 in column 1, the prior influences the training only in the initial 10 epochs. After epoch 10, the gradient ratio is around the same scale or smaller.

## A.5 DETAILS OF HEATMAP SHRINKAGE

To find out whether the added prior influences the spread of the heatmaps, we investigate the shrinkage change of heatmaps by separating the samples. We separate the samples based on easy/medium versus hard cases, where the former generally produce localized heatmaps while the latter have dispersed heatmaps as per Fig. I; the alternative is to separate by thresholding the activation sum in a fixed area (Eq. 6). We consider the activation sum $A(1) > 0.34$ of the integral regression baseline as localized, otherwise as dispersed (threshold 0.34 selected based on Fig. I). We tally the activation sums for the localized and dispersed heatmaps, without and with the prior for different values of $\sigma$. Activation sums $A(s)_{e/m}$, $A(s)_h$, $A(s)_l$, $A(s)_h$, and Average Precision $\text{AP}_{e/m}$, $\text{AP}_h$, $\text{AP}_l$, $\text{AP}_h$ indicate the extent of localization and the performance of the two conditions, respectively. An increase in activation sum indicates further heatmap localization; a decrease indicates dispersion. The "No prior" row is the baseline integral pose regression.

From Table IV (a), both separations present the same trend. For localized cases, forcing $\sigma$ to be degenerately small or too large ($\sigma = 0.5$ or $\sigma = 4$) increases/decreases the activation sum over the baseline, thereby confirming the effect of the prior as intended. The change in AP follows, *i.e.* decreases when $\sigma$ becomes incompatible from being too small or too large. For dispersed cases, the prior shrinks the activation sum as expected. Like the localized case, there is an optimal range for which improvement in AP is better; degenerate $\sigma$ will decrease the AP from the baseline.

| $\sigma$ | regression | | | | | detection | | | | | |
|---|---|---|---|---|---|---|---|---|---|---|---|
| $s$ | 0.5 | 1 | 2 | 3 | 4 | 1 | 2 | 2.5 | 3 | 3.5 | 4 |
| 4 | >1000 | 3.24 | 3.53 | 3.75 | 3.89 | >1000 | 2.52 | 0.64 | 0.72 | 0.85 | 0.88 |

Table III: Results of Pearson Chi-square test between distribution of heatmap $\mathbf{H}$ and Gaussian templates $\mathbf{T}$ with different window size $s$ and standard deviation $\sigma$ in easy cases. Accordingly, for detection, $\sigma$ should be approximately 2.5; while for regression, optimal $\sigma$ should be between 0.5 and 1.

| $w_p : \lambda w_i$ | 100:1 | 10:1 | 1:1 | 1:10 | 1:100 |
|---|---|---|---|---|---|
| $+\sigma=0.5$ | 63.6 | 64.8 | 65.1 | 65.7 | 66.4 |
| $+\sigma=1$ | 67.6 | 68.1 | 68.5 | 68.3 | 67.8 |
| $+\sigma=2$ | 68.5 | 70.8 | 71.4 | 70.6 | 67.8 |
| $+\sigma=3$ | 68.2 | 69.4 | 69.8 | 69.2 | 68.1 |
| $+\sigma=4$ | 64.2 | 66.1 | 66.9 | 66.9 | 67.0 |

(a)

| $w_p : \lambda w_i$ | 100:1 | 10:1 | 1:1 | 1:10 | 1:100 |
|---|---|---|---|---|---|
| epoch 1 | 2 | 2 | 2 | 1 | 1 |
| epoch 10 | 2 | 1 | 0 | 0 | 0 |
| epoch 30 | 2 | 0 | 0 | 0 | -1 |
| epoch 50 | 1 | -1 | -1 | -1 | -1 |
| epoch 80 | 0 | -1 | -1 | -1 | -2 |

(b)

Table IV: (a) Performance of the integral model adding different priors with different weight ratios $w_p : \lambda w_i$. The performance depends largely on different $\sigma$s, but less influenced with reasonable change. (b) Ratio of the gradient magnitudes of combined model (integral+$\sigma = 2$) with different weight ratios $w_p : \lambda w_i$. The gradient ratio decreases over the course of training. After epoch 10, the gradient ratio is around the same scale or smaller.

| model | $A(2)_{e/m}$ | $AP_{e/m}$ | $A(2)_{e/m}$ | $AP_l$ | $A(2)_h$ | $AP_h$ | $A(2)_l$ | $AP_l$ |
|---|---|---|---|---|---|---|---|---|
| No prior | 0.65 | 70.4 | 0.71 | 72.3 | 0.35 | 44.7 | 0.33 | 38.9 |
| $+\sigma=0.5$ | 0.75 | 68.2 | 0.79 | 69.8 | 0.36 | 43.9 | 0.36 | 35.2 |
| $+\sigma=1$ | 0.63 | 71.9 | 0.70 | 73.9 | 0.36 | 45.2 | 0.31 | 39.1 |
| $+\sigma=2$ | 0.54 | 74.2 | 0.59 | 76.1 | 0.41 | 47.4 | 0.37 | 39.8 |
| $+\sigma=3$ | 0.49 | 73.0 | 0.56 | 74.9 | 0.39 | 47.6 | 0.36 | 39.3 |
| $+\sigma=4$ | 0.45 | 70.2 | 0.55 | 71.7 | 0.39 | 46.9 | 0.38 | 39.5 |

Table V: Localization extent and performance of divided localized samples versus dispersed samples. The extent of localization largely depends on the added prior. With an optimal $\sigma = 2$ chosen, the previously localized heatmaps have been dispersed with improved performance.

## A.6 DERIVATION OF EQ. 10

Eq. 10 is obtained by the chain rule. Therefore, we list the value of each term as follows:

$$\frac{\partial L_{\text{re}}}{\partial \hat{\mathbf{J}}} = \begin{bmatrix} s(\hat{J}_x - J_x) \\ s(\hat{J}_y - J_y) \end{bmatrix}, \quad \frac{\partial \hat{\mathbf{J}}}{\partial \tilde{h}_{\mathbf{p}}} = (i+j), \quad \frac{\partial \tilde{h}_{\mathbf{p}}}{\partial \hat{h}_{\mathbf{q}}} = \begin{cases} \beta \tilde{h}_{\mathbf{q}}(1 - \tilde{h}_{\mathbf{q}}), & \mathbf{q} = \mathbf{p} \\ -\tilde{h}_{\mathbf{p}} \tilde{h}_{\mathbf{q}}, & \mathbf{q} \neq \mathbf{p} \end{cases}, \quad \text{(V)}$$

where $\mathbf{q} = (u, v)$ can be any pixel in $\Omega$. Combining the terms, gradients at $\mathbf{p}$ can be calculated by

$$\frac{\partial L_{\text{re}}}{\partial \hat{h}_{\mathbf{p}}} = \beta((\hat{J}_x - J_x)i + (\hat{J}_y - J_y)j)\tilde{h}_{\mathbf{p}}(1 - \tilde{h}_{\mathbf{p}}) - \sum_{\substack{\mathbf{q} \in \Omega, \\ \mathbf{q} \neq \mathbf{p}}} \beta((\hat{J}_x - J_x)u + (\hat{J}_y - J_y)v)\tilde{h}_{\mathbf{q}}\tilde{h}_{\mathbf{p}}. \quad \text{(VI)}$$

Eq. 10 can be obtained by rearranging the terms of Eq. VI.

## A.7 DERIVATION OF EQ. 11

Let us suppose $\mathbf{H}_n = g(\mathbf{w}_n | I)$ where $\mathbf{H}_n$ is the heatmap of a given image $I$ at the $n$-th iteration, $g(\mathbf{w}_n)$ is the function of generating the heatmap, *i.e.* the backbone network, with weights at the $n$-th iteration. According to a Taylor expansion, we arrive at

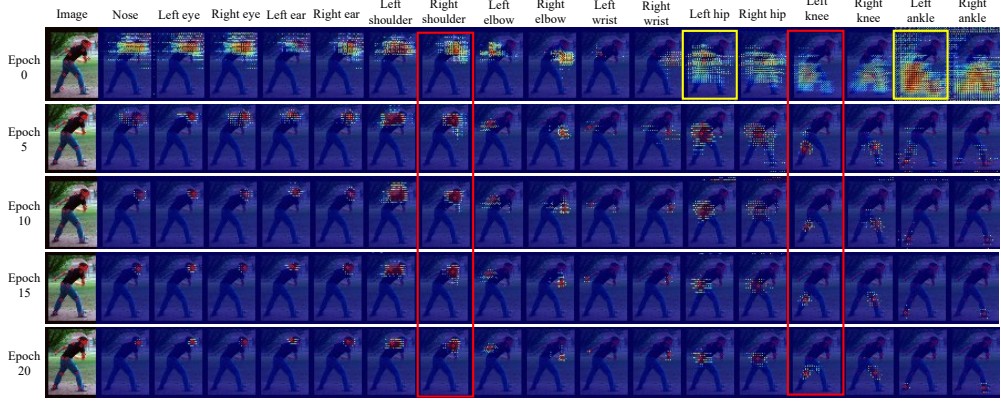

Figure II: Visualization of the training performance of integral pose regression on one single image at epoch 0, 5, 10, 15, 20 (from top to down). Corner or edge pixels of the correct quadrant are largely activated in the epoch 0 (highlighted in the yellow boxes). Further training is focused on narrowing down the activated region (shown in the red boxes).

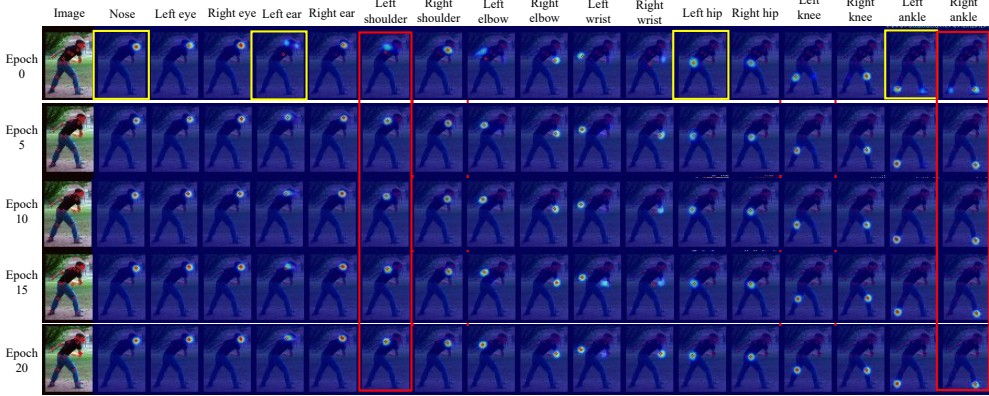

Figure III: Visualization of the training performance of detection-based methods on one single image at epoch 0, 5, 10, 15, 20 (from top to down). One epoch has already enabled the network to roughly localize the joint compared with regression-based method (in the yellow boxes). Further training of detection-based methods makes the predictions more stable and precise (highlighted in red boxes).

$$g(\mathbf{w}_{n+1}) = g(\mathbf{w}_n) + \frac{g^{'}(\mathbf{w}_n)}{1!}(\mathbf{w}_{n+1} - \mathbf{w}_n) + \frac{g^{''}(\mathbf{w}_n)}{2!}(\mathbf{w}_{n+1} - \mathbf{w}_n)^2 + \dots, \qquad \text{(VII)}$$

where $\mathbf{w}_{n+1} - \mathbf{w}_n = -\alpha\nabla g(\mathbf{w}) = -\alpha\frac{\partial L}{\partial \mathbf{H}}g^{'}(\mathbf{w}_n)$ according to gradient descent, and $(\mathbf{w}_{n+1} - \mathbf{w}_n)^i$ is negligible compared with $(\mathbf{w}_{n+1} - \mathbf{w}_n)$ when $i > 1$. Therefore, Eq. VII can be rewritten as

$$\mathbf{H}_{n+1} = g(\mathbf{w}_{n+1}) \approx g(\mathbf{w}_n) + g^{'}(\mathbf{w}_n)(\mathbf{w}_{n+1} - \mathbf{w}_n) = \mathbf{H}_n - \alpha\frac{\partial L}{\partial \mathbf{H}}||g^{'}(\mathbf{w}_n)||^2. \qquad \text{(VIII)}$$

Therefore, for each pixel $h_{\mathbf{p}}$ in $\mathbf{H}$, the update rule is

$$h_{\mathbf{p}}^{n+1} \approx h_{\mathbf{p}}^{n} - \gamma\nabla_{\mathbf{p}}^{n}, \qquad \text{(IX)}$$

where $\gamma$ represents $\alpha||g^{'}(\mathbf{w}_n)||$, $h_{\mathbf{p}}^{n}$ and $\nabla_{\mathbf{p}}^{n}$ denote heatmap and gradient value located at $\mathbf{p}$ in the heatmap at the $n$th iteration. $h_{\mathbf{p}}^{0}$ denotes the original heatmap.

## A.8    REALWORLD VISUALIZATIONS

We show the visualization of the training performance of integral pose regression and detection-based methods on one single image in Figs. II and III, respectively. We can see that the further training of integral pose regression is focused on narrowing down the activated region while the further training of detection-based methods makes the predictions more stable and precise.

