# OpenReview forum: "Dive Deeper Into Integral Pose Regression"
_ICLR.cc/2022/Conference — ICLR 2022 Poster_

### Official Review · Reviewer_hXfD · 2021-11-02

**Correctness:** 2
**Technical Novelty And Significance:** 3
**Empirical Novelty And Significance:** 2
**Recommendation:** 6
**Confidence:** 5

**Main Review:**

Pros:
1. This paper gives an interesting hypothesis on the cause of performance inconsistency in IPR methods.
2. Some theoretical evidence and empirical results are provided to help understand and verify this hypothesis.

Cons:
1. The authors claim that extremely localized heatmaps degrade performance (of IPR methods on easy samples). But this claim is not well supported by empirical results or theory.

    (1) To verify the impact of sigma on IPR methods, the authors introduce a distribution prior to prevent the shrinkage of the heatmap. However, this method may also introduce some other confounding factors. For example, what are the weights of the prior loss term and the integral loss term? What are the magnitudes of the gradients produced by these two loss functions? Would the prior loss term dominate the training process thus Table 1 mainly reflects the impact of sigma on detection methods? These are not revealed.

   (2) As shown by Fig 2(b), not all samples converge to the extremely localized heatmaps at the end of the training. After introducing an appropriate sigma prior, the overall performance improves as in Table 1. However, the respective impact of sigma on the previously localized heatmaps and previously dispersed heatmaps are not clearly shown. It might be better if the authors would show the change of shrinkage and performance separately on these two kinds of samples.

   (3) Although theoretical evidence is provided to help understand the behavior of gradients of IPR methods, no theoretical evidence is provided to directly support the main claim: extremely localized heatmaps degrade performance.

2. The novelty of the proposed heatmap prior loss is limited. It has already been explored in the initial IPR paper [1]. The authors in [1] combine the heatmap loss with the integral loss and show that this improves over the pure integral loss. What is more, the proposed heatmap prior loss does not substantially improve the gradient behavior of the IPR process. It is more likely that the prior loss dominates the backward update. It might be better if the authors would give more details.

Minor Comments:

1. Abbreviation was used before detailed explanation.  e.g.  Introduction Section :  EPE
2. Typos:
    (1) Subsection 3.3  hte --> the
    (2) Subsection 4.2  E_{de} <= E_{re} -->  E_{de} >= E_{re}
    (3) Subsection 5.1  location factor --> value factor
3. Subsection 5.1: The detailed derivation of Eq10 is not shown in the Supplementary.

[1] Sun et al. Integral human pose regression. ECCV 2018


**Summary Of The Paper:**

This paper investigates the performance inconsistency of integral pose regression (IPR) methods on 'easy' pose samples and 'hard' pose samples (v.s. argmax methods). The authors find that the heatmap shrinkage caused by the IPR methods leads to the lower accuracy on 'easy' samples than argmax based methods. They give some toy examples and intuitively demonstrate how the shrinkage of heatmaps is caused by the gradient of the IPR method. They further propose a heatmap distribution prior loss to mitigate the shrinkage.

**Summary Of The Review:**

Because the main claims are not well supported and the proposed heatmap prior loss lacks novelty, I lean towards reject.

---

> ### Author Response · Authors · 2021-11-22
> **Thanks and response to concerns (p1)**
>
> Thank you for the careful review and constructive feedback.
>
> Our paper has two key parts: Section 4 on Heatmap decoding and Section 5 on Supervision and learning. We kindly request that Section 5 not be overlooked when evaluating the contribution of our work, as it helps understand the learning of integral method by investigating gradients.
>
> In regards to the **novelty of the heatmap prior loss**, we strongly disagree that this should be regarded as a "con" of our work:
> * We don't claim the prior loss as our novelty, nor as one of our contributions.
> * We use the prior loss merely as an experimental setting to further verify our hypothesis on the localization heatmaps.
> * We are happy to add mention of [a][b][c] who use similar priors in their work.
>
> We answer specific questions below:
>
> ---
>
> **Q1.1: "distribution prior introduces cofounding factors, e.g. from weighting of the loss terms"**
>
> **A1.1:** Thanks for raising this insightful point. The short conclusion is the influence of weights of two losses is not large and the reason is that the gradients shrink from the prior as training progresses. We provide analysis and experiments below and also in the revised paper.
>
> To verify, we add an ablation to check the impact loss weightings on the AP (Table 1) and magnitude of the gradients (Table 2).
>
> Table 1
>
> $w_p:\lambda w_i$ | 100:1 | 10:1 | 1:1 |1:10| 1:100
> ---|---|---|---|---|---
> +$\sigma$=0.5|63.6|64.8|65.1|65.7|66.4
> +$\sigma$=1|67.6|68.1|68.5|68.3|67.8
> +$\sigma$=2|68.5|70.8|71.4|70.6|67.8
> +$\sigma$=3|68.2|69.4|69.8|69.2|68.1
> +$\sigma$=4|64.2|66.1|66.9|66.9|67.0
>
>
>
> $w_p$ and $w_i$ are the weights of the prior and integral losses respectively. For intuitive comparison, we consider the ratio $w_pL_p : w_i\cdot \lambda L_i$ where $\lambda $=10$^{-2}$ to set the two gradients $L_p$ and $L_i$ to approximately equal magnitudes when $w_p = w_i$.
>
> Table 1 shows that the AP depends heavily on the value of $\sigma$ but for some fixed $\sigma$ (within each row), the ratio of loss weightings has less impact and results are stable within 1-2$\%$,with larger differences only at the extreme settings of 100:1 or 1:100.  The ratio of gradient magnitudes in Table 2 helps explain why the weightings do not have much impact:
>
> Table 2
>
> $w_p:\lambda w_i$ | 100:1 | 10:1 | 1:1 |1:10| 1:100
> ---|---|---|---|---|---
> epoch 0| 2 | 2 | 2 | 1 | 1
> epoch 10| 2 | 1 | 0 | 0 |0
> epoch 30| 2 |0|0|0|-1
> epoch 50|1| -1 |-1|-1|-1
> epoch 80|0|-1|-1|-1|-2
>
> Table 2 gives the mean ratio $\frac{g_p}{g_i}$ where $g_p$ and $g_i$ are the gradient magnitutes of the prior loss and integral loss respectively over the course of training. A column of 2/1/0/1/-2 means 10$^{2}$/10$^{1}$/10$^{0}$/10$^{-1}$/10$^{-2}$ ratio in gradients over the epochs.
>
> Table 2 shows that regardless of the weighting, the gradient ratio decreases over the coruse of training (i.e. dropoff within each column).  Apart from an extremely high weighting of the prior, i.e. 100:1 in column 1, the prior influences training only in the initial 10 epochs. After epoch 10, the gradient ratio is around the same scale or smaller. Therefore, the prior loss influences the network at initial epochs, while the integral loss dominates afterwards.
>
> ---
>
>
> [a] Sun et al. Integral pose regression. ECCV 2018.
>
> [b] Gu et al. Removing the bias of integral pose regression. ICCV 2021.
>
> [c] Nibali et al. Numerical coordinate regression with convolutional neural networks. arXiv:1801.07372 (2018).

---

> > ### Author Response · Authors · 2021-11-22
> > **Thanks and response to concerns (p2)**
> >
> > **Q1.2: "what's the respective impact of localized and dispersed heatmap after introducing a sigma prior?"**
> >
> > **A1.2:** Thank you for the suggestion of separating the two cases. The shortened answer is that after separating localized and dispersed samples with two criteria, by adding proper distribution prior (e.g., $\sigma$=2), both results suggest that the previously localized heatmaps become more dispersed and performance is improved meanwhile. The detailed analysis and experiments are introduced below and also included in the revised paper.
> >
> > We can separate either based on easy/medium cases (localized) vs. hard (dispersed); an alternative is to separate by thresholding the activation sum in a fixed area (Eq. 6 in main paper). We consider activation sums $A(1)$ > 0.34 as localized, otherwise as dispersed (threshold selected based on Table 5 of Supplementary).
> >
> >
> > We tally the activation sums for the localized vs. dispersed heatmaps without the prior vs. adding priors with various sigma.
> > Activation sums $A(s)_l$, $A(s)_d$ and Average Precision AP$_l$ and AP$_d$ indicate the extent of localization and performance of the two conditions. An increase in activation sum indicates further heatmap localization; a decrease indicates dispersion.  The "No prior" row is the baseline integral pose regression.
> >
> > Table 3 - easy/medium (localized) vs. hard (dispersed)
> >
> > |model| $A(1)_{e/m}$| $A(1)_d$|AP$_{e/m}$|AP$_d$
> > ---|---|---|---|---
> > No prior|0.37|0.20|70.4|44.7
> > +$\sigma$=0.5|0.53 (+0.16)|0.24 (+0.04)|68.2 (-2.2)|43.9 (-0.8)
> > **+$\sigma$=2**|**0.26 (-0.11)**|**0.25 (+0.05)**|**74.1 (+3.7)**|**47.4 (+2.7)**
> > +$\sigma$=4|0.21 (-0.15)|0.24 (+0.04)|70.2 (-0.2)|46.9 (+1.8)
> >
> > Table 4 - localized ($A(1) < 0.34$) vs. dispersed (otherwise)
> >
> > |model| $A(1)_l$| $A(1)_d$| AP$_l$|AP$_d$
> > ---|---|---|---|---
> > No prior|0.44|0.18|72.3|38.9
> > +$\sigma$=0.5|0.58 (+0.14)|0.19 (+0.01)|69.8 (-2.5)|35.2 (-3.7)
> > **+$\sigma$=2**|**0.31 (-0.13)**|**0.21 (+0.03)**|**76.1 (+3.8)**|**39.8 (+0.9)**
> > +$\sigma$=4|0.25 (-0.19)|0.22 (+0.04)|71.7 (-0.6)|39.5 (+0.6)
> >
> > For localized cases, forcing $\sigma$ to be degenerately small or too large ($\sigma$ = 0.5 or $\sigma$ = 4) increases/decreases the activation sum wrt the baseline, thereby confirming the effect of the prior as intended.  The change in AP follows, i.e. decrease when $\sigma$ incompatible from being too small or too large.
> >
> > For dispersed cases, the prior shrinks the activation sum as expected. Like the localized case, there is an optimal range for which improvement in AP is better ($\sigma$ = 2, 4); degenerate $\sigma$ will decrease the AP from the baseline ($\sigma = 0.5$).
> >
> > ---
> >
> > **Q1.3: "no theoretical evidence for the claim, extremely localized heatmaps degrade the performance?"**
> >
> > **A1.3:** Having theoretical support is of course ideal and we will look into this for future work.
> >
> > We openly state in Findings/Contribution item 2 that localized heatmaps is a hypothesis derived from empirical evidence. We are happy to adjust our language to further emphasize this point. We arrive at the hypothesis from three experiments:
> >
> > *   imposing an extremely small vs. appropriate $\sigma$ (e.g., $\sigma$=0.5/1 vs. $\sigma$=2) worsens detection methods' results
> > *   heatmaps for easy cases are more localized for integral regression than detection methods; when fitted with a Gaussian, regression methods exhibit a smaller $\sigma$ than detection methods ([0.5, 1] vs. $>2$, see main paper Fig. 2c)
> > *   imposing a prior with larger $\sigma$ ($\sigma$=2) is correlated with improved performance (main paper Table 1); from the ablation of **A1.2** (Tables 3 &4) we conclusively show that priors with larger does indeed increase the heatmap localization for the easy cases / localized heatmaps
> >
> > **Q2: "prior loss does not substantially improve the gradient behavior. It's morely likely that prior loss gradients dominate in the backward."**
> >
> > **A2:** This statement is not clear as we do not make any claims in regards to the prior loss with respect to the gradient behaviour.  Our main purpose of adding the prior is simply to verify our hypothesis on the localized heatmaps.
> >
> > Table 2 in **A1.1** shows definitively that the prior loss gradients **do not** dominate the backwards update.

---

> > ### Comment · Reviewer_hXfD · 2021-11-25
> > **Response to Rebuttal (p1)**
> >
> > Thanks for the authors' response and efforts. These experiments address my concerns. I decide to raise my rating to WA.

---

### Official Review · Reviewer_mXxk · 2021-11-02

**Correctness:** 4
**Technical Novelty And Significance:** 2
**Empirical Novelty And Significance:** 3
**Recommendation:** 6
**Confidence:** 4

**Main Review:**

- The paper considers a very interesting empirical setting. Keypoint Localization is a very popular task and many approaches rely on this. Getting more insight about each different method can be valuable to many people.

- The paper combines some theoretical exploration along with empirical validation/demonstration in both toy settings, but also real settings. I find very useful this combination of theory/experiments and toy examples/real world settings. For example, the toy experiment shows different behavior than the real world training, so it is important to look at both of them at the same time. On the contrary, the theoretical study is a little basic, but it is enough to demonstrate the insights of the authors.

- The insights and trade offs discussed are very useful for practitioners that use these methods. Some of these observations also exist in previous works, but it is helpful to have papers like that, that collect this type of observations (e.g., trade off for training speed, convergence, potential vanishing gradients, etc)

- I really appreciate that when the authors are making specific assumptions (i.e., about the form of the heatmaps), they also provide the empirical demonstration of the fact.

- In figure 4, the top red window (in third frame of bottom row) is not clear that it corresponds to the zoomed in version of the bottom window. I thought its goal was to indicate activations in locations other than the ankle. It would be good to indicate this to avoid confusion.

- Is there any exploration to be done for cases where these two approaches are combined? E.g., training with heatmap supervision in the beginning to warm up the network, and with integral supervision later to get the best performance for hard examples too?

**Summary Of The Paper:**

This paper is an in-depth study of the Integral Pose Regression method that is very common for Keypoint Localization. Integral Pose Regression has been adopted by many methods, and it is probably one of the two main paradigms for Keypoint Localization, along with Heatmap Regression. Many works have compared the two approaches, and in some cases (e.g., Gu et al, 2021) they provide a more detailed empirical comparison, identifying the strong and weak points of each approach. This paper has a similar goal, but combines theoretical investigation with empirical results to highlight insights about the two methods.

**Summary Of The Review:**

Overall, I am positive about this paper. Obviously, the novelty is limited, but I believe that its value is high, particularly for practitioners. Since the analysis done here is more basic, I will give it a Weak Accept rating, but I think it is important to accept it to encourage similar efforts.

---

> ### Author Response · Authors · 2021-11-22
> **Thanks and response to concerns**
>
> Thank you for the careful review and constructive suggestions.
>
> Our paper collects some observations from previous works ([a][b][c]) but several are also our own findings which are perhaps known but not mentioned in the literature:
> *   Degenerately small regions of activation degrade the accuracy of both detection and integral methods.
> *   The integral regression method suffers from gradient vanishing and provides less spatial cues for learning the heatmap, making the learning less efficient than detection-based methods.
>
>
> Our theoretical analysis on the decoding and gradients is a first foray into understanding integral pose regression. Each assumption and conclusion is also grounded by experimentation. While one can construct more sophisticated formalism, this complicates efforts to link the theory to the empirical evidence. We want to stress that such an analysis is the first of its kind in pose estimation, which we believe also has merits for novelty.
>
> For the posed questions:
>
> ---
>
> **Q1: Fig. 4 "the top red window is not clear?"**
>
> **A1:** Thanks for this suggestion.  We revised this figure to improve clarity. This zoomed in region illustrates that activations contain very few pixels and that the activations are not precisely located at the ankle.
>
> ---
>
> **Q2: “Exploration ... for cases where two approaches are combined?”**
>
> **A2:** A few works have considered combining the two approaches though conclusions are difficult to draw at the moment. [a] adds the per-pixel loss to integral regression and found that this has benefits primarily for 3D (but not 2D) pose estimation. [b] uses a per-pixel loss to warm up network training and then switched over to integral supervision, **as suggested in the comment**. but shows only experimental demonstration. [c] tested the use of different priors on the heatmap and found that KL divergence is the most suitable.  These works differ from our aims in exploring how the spatial spread of the heatmap will influence performance.
>
> We thank the reviewer for the suggestion and will think further about the insights of combining the two methods and plan this for future work.
>
> [a] Sun et al. Integral pose regression. ECCV 2018.
>
> [b] Gu et al. Removing the bias of integral pose regression. ICCV 2021.
>
> [c] Nibali et al. Numerical coordinate regression with convolutional neural networks. arXiv:1801.07372 (2018).

---

> > ### Comment · Reviewer_mXxk · 2021-11-29
> > **Response to Rebuttal**
> >
> > I want to thank the authors for their response. They have addressed well my questions and comments. I think this paper should get accepted and I'm keeping my Weak Accept rating.

---

### Official Review · Reviewer_oYhA · 2021-11-03

**Correctness:** 3
**Technical Novelty And Significance:** 3
**Empirical Novelty And Significance:** Not applicable
**Recommendation:** 8
**Confidence:** 4

**Main Review:**

## Strengths

- The paper is well written and organized.
- A theoretical study on how these two pose estimation methods, detection-based and integral regression-based, work is missing in the literature. This work has well studied this.
- Comprehensive experiments are conducted to support the theoretical analysis.


## Weaknesses
- How to apply the conclusion in this work to further improve the pose estimation is missing. This is not a big issue to me. However, it would be great if the author can give such a guide according to the theoretical analysis.

**Summary Of The Paper:**

Heatmap detection-based method and integral regression-based method are widely used in the pose estimation task. The paper gave a detailed comparison and theoretical analysis between these two methods. A unified model is proposed to compare these two methods. Comprehensive experiments are conducted to support the theoretical analysis.

**Summary Of The Review:**

In the field of human pose estimation, the heatmap detection-based method and integral regression-based method are widely used. Previous works showed the performance and behavior differences between these two approaches. However, there are rarely works to theoretically study where the differences come from. This work proposed a unified model to analyze these two methods and is well supported by comprehensive experiments. It's great to see such a theoretical study on that. I'd like this paper, and I believe this work can help the research community in the field of human pose estimation to better understand the problem. Though the novelty of the paper is limited, I think we should encourage such efforts on how to theoretically analyze our methods.

---

> ### Author Response · Authors · 2021-11-22
> **Thanks and response to concerns**
>
> Thank you for the careful review and constructive suggestions.
>
> In this paper, we offer our understanding and insights of the integral regression vs. heatmap detection methods used for human pose estimation.  We revised our conclusions to orient it more towards improving pose estimation. Currently, our findings suggest that the use of prior with an appropriate spatial spread helps the end-to-end integral regression.

---

> > ### Comment · Reviewer_oYhA · 2021-11-30
> > **Response to rebuttal**
> >
> > I keep my recommendation.

---

### Author Response · Authors · 2021-11-23
**Thanks for the reviews and summary of key paper changes**

We thank the reviewers for their careful and thoughtful reviews. The reviewers appreciated our theoretical analysis of the difference between detection and integral methods (R1, R2, R3), comprehensive experiments to support the analysis (R1, R2), and insights to better serve the pose estimation community (R1, R2).

Since the concerns voiced by the reviewers are mainly non-overlapping, we offer detailed responses individually to each review.

The key changes to the paper can be summarized as follows:

* revised the conclusion to orient it more towards improving pose estimation performance.
* clarified Fig. 4.
* added analysis and experiments about different weights and gradient magnitudes of prior loss and integral loss in Sec. 4.2 and Appendix A.4.
* added experiments in Table 1 regarding performance and activation sum of easy/medium cases and hard cases.
* fixed some typos.

---

### Decision · Program_Chairs · 2022-01-20

**Decision:**

Accept (Poster)

**Comment:**

All reviewers recognized the contribution of performing a theoretical study to investigate how the two technique lines (detection-based methods and integral regression-based methods) work for pose estimation. The study in this paper could be valuable for the researchers in the pose estimation domain, for the further research. The AC agrees with the reviewers and recommends accept for this paper.